# Off-target effects of CRISPRa on interleukin-6 expression

**Sébastien Soubeyrand**[1]*, **Paulina Lau**[1], **Victoria Peters**[1], **Ruth McPherson**[1,2]*

**1** Atherogenomics Laboratory, University of Ottawa Heart Institute, Ottawa, Ontario, Canada, **2** Department of Medicine, Ruddy Canadian Cardiovascular Genetics Centre, University of Ottawa Heart Institute, Ottawa, Ontario, Canada

* ssoubeyrand@ottawaheart.ca (SS); rmcpherson@ottawaheart.ca (RM)

**Data Availability Statement:** Transcriptome profiling on SG-286/CRISPRa transfected cells was performed at the Centre for Applied Genomics (The Hospital for Sick Children, ON, Canada). Purified RNA, was converted to cRNAs and hybridized to

## Abstract

Inactive fusion variants of the CRISPR-Cas9 system are increasingly being used as standard methodology to study transcription regulation. Their ability to readily manipulate the native genomic loci is particularly advantageous. In this work, we serendipitously uncover the key cytokine *IL6* as an off-target of the activating derivative of CRISPR (CRISPRa) while studying *RP11-326A19.4*, a novel long-non coding RNA (lncRNA). Increasing *RP11-326A19.4* expression in HEK293T cells via CRISPRa-mediated activation of its promoter region induced genome-wide transcriptional changes, including upregulation of *IL6*, an important cytokine. *IL6* was increased in response to distinct sgRNA targeting the *RP11-326A19.4* promoter region, suggesting specificity. Loss of the cognate sgRNA recognition sites failed to abolish CRISPRa mediated activation of *IL6* however, pointing to off-target effects. Bioinformatic approaches did not reveal predicted off-target binding sites. Off-target activation of *IL6* was sustained and involved low level activation of known *IL6* regulators. Increased *IL6* remained sensitive to further activation by TNFα, consistent with the existence of independent mechanisms. This study provides experimental evidence that CRISPRa has discrete, unpredictable off-targeting limitations that must be considered when using this emerging technology.

## Introduction

Clustered Regularly Interspaced Short Palindromic Repeats (CRISPR) related technologies are revolutionizing gene editing. Not only do CRISPR-associated (Cas) proteins allow for relatively facile genomic editing, inactive variants are continuously being created to study gene function [1,2]. Specificity of Cas9 has been extensively studied [3]. Optimal cleavage by *sp*Cas9 requires anRNA co-factor, either a two-part guide RNA component (native context) or in a more artificial and practical setting, a single-guide RNA (sgRNA) consisting of a 20-mer spacer sequence that confers cleavage specificity and a longer generic scaffold. In addition, the cognate DNA sequence must be followed by a NGG protospacer adjacent motif (PAM). This requirement allows, in theory, spCas9 to cut at a unique site of the human genome. In practice however, cleavage can occur at other sites [3,4]. Furthermore, variants harnessing the DNA

Human Clariom D arrays. Results were analyzed by using the Expression and Transcriptome Analysis Consoles (Applied Biosystems); Array data have been deposited at the GEO repository as GSE132451.

**Funding:** This work was funded by a Canadian Institutes for Health Research Foundation grant (RM). The funders had no role in study design, data collection and analysis, decision to publish, or preparation of the manuscript.

**Competing interests:** The authors have declared that no competing interests exist.

homing properties of Cas proteins, and in particular Cas protein 9 (Cas9), are continuously being engineered to be used as research tools. The resulting fusion proteins are typically comprised of a deactivated Cas9 (dCas9) that retains its DNA binding properties, fused to various regulatory domains that can influence genomic regulation, including transcriptional activation and repression (CRISPRa and CRISPRi, respectively). Compared to wild-type Cas9, dCas9 fusions need only to bind DNA in the proper orientation to impart *de novo* function. Current evidence indicates that each sgRNA has the potential to direct Cas9 or dCas9 to tens to thousands of sites [5,6]. While this represents an upper limit and *de facto* binding events are likely not as pervasive due to nucleosome interference [7], the genome contains abundant potential off-target binding sites. Whether these binding events persist long enough to have functional outcomes is unclear.

Driven by increasingly sensitive genome-wide expression analyses, a new level of regulatory control has emerged over the last decade, as it was realized that a significant proportion of the genome which was heretofore believed to be transcriptionally silent was rather expressing long transcripts. Although they lack salient protein coding potential, these long non-coding RNAs (lncRNA) form a class of RNAs in many respects similar to traditional mRNA: lncRNA are relatively long, contain polyA tails and undergo alternative splicing, although splicing of lncRNA is typically more extensive [8]. In addition conservation tends to be lower than for protein coding genes [9,10]. Some but not all lncRNA appear to be functional [11], which may reflect our inability to properly test for biological significance [12]. To account for their lack of conservation, it has been proposed that the very act of transcription at lncRNA loci, rather than the transcript *per se*, determines function [13]. With these features in mind, the advent of CRISPRa methodology seemed particularly suited to tackle lncRNA function as targeting CRISPRa/i upstream of lncRNA genes in principle 1- permits transcriptional activation (or repression) and 2- induces changes in the native transcript levels, including all forms of spliced variants, whose ratio may be important for function.

IL6 is a broadly expressed cytokine with wide ranging impacts on multiple biological systems. Key implications of IL6 in cancer, inflammation, coronary artery disease (*IL6R*), hematopoiesis, and metabolic processes have been reported [14–17]. Using different binding modalities in specific contexts, IL6 provides either pro- or anti-inflammatory cues. A considerable body of literature dealing with regulation of *IL6* has shown it to be under extensive transcriptional and translational control. Post-transcriptionally, *IL6* is a relatively unstable transcript, regulated by several microRNA and protein regulators that interact with the 3' UTRs of the mature transcript [15].

Here we demonstrate that targeting CRISPRa to distinct regions of a novel lncRNA promoter region leads to *IL6* upregulation, largely via off-target effects. The mechanisms underpinning the activation implicate increased transcription via a low level activation of established *IL6* inducers. These findings add a note of caution regarding the reliability of current CRISPRa technology.

## Results

### Validation of a long non-coding RNA situated proximal to a region associated with coronary artery disease

While investigating the role of a GWAS-identified locus for Coronary Artery Disease (CAD) at15q26 near *MFGE8* [18], we identified a region encoding putative, uncharacterized transcripts (*RP11-326A19.4/TCONS_00023492*) immediately adjacent to the risk region (Fig 1A). The proximity of the transcripts to the CAD region hinted at a possible link to

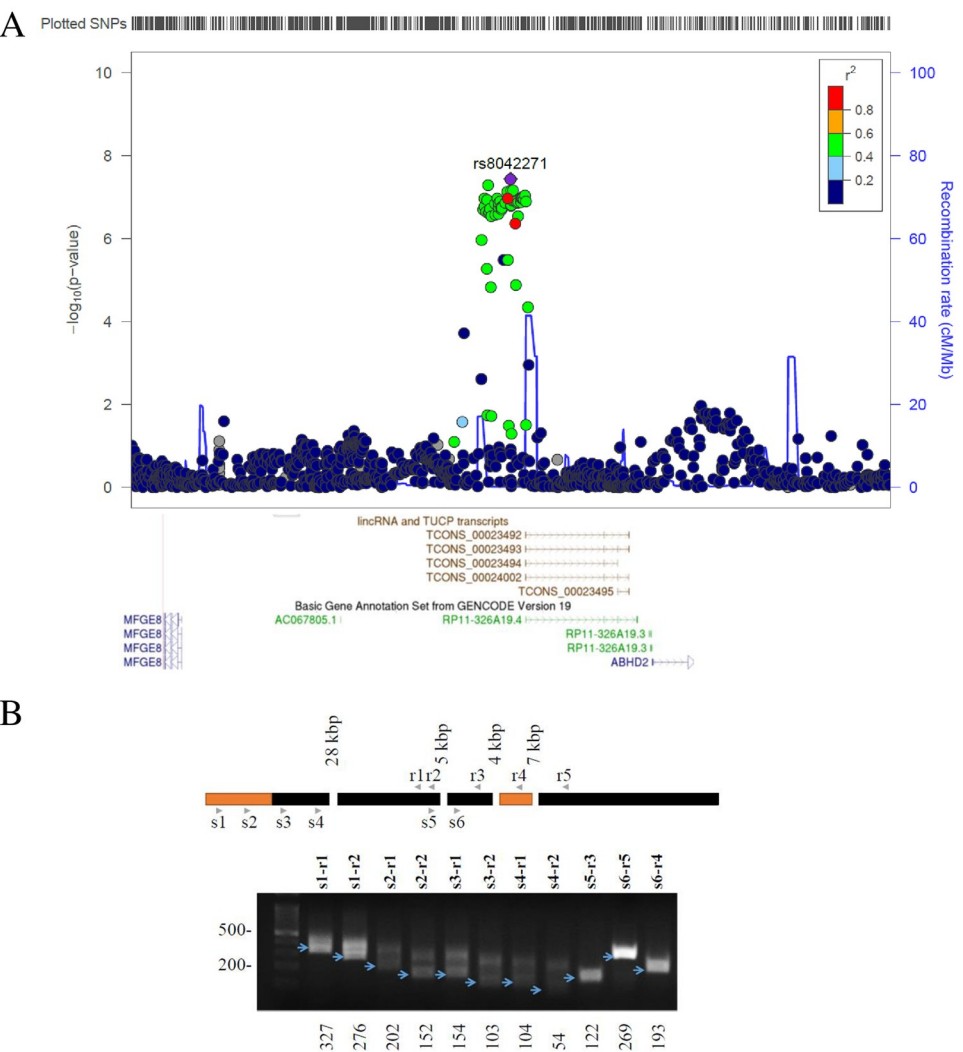

**Fig 1. Validation of *RP11-326A19.4/TCONS* forms in HEK293T.** A, Manhattan plot of the 15q.26 CAD risk locus. Top, local Manhattan plot Data was drawn with LocusZoom using CAD association data from Nikpay et al [34]. Bottom corresponding genes and transcripts from GENECODE (green and blue) and Human Body Map (brown), visualized via the UCSC browser data, to scale with the Manhattan plot. B, splice form analysis of *RP11-326A19.4*. RNA was isolated from HEK293T and subjected to qRT-PCR analyses with primer sets targeted against various exons of the predicted transcripts. Top, schematic of primers used for PCR and alternating exon organization. Black bars represent exons shared by both spliced forms while orange bars represent additional exonic sequences unique to the TCONS_00023493 form. Size of predicted introns are indicated on top. Bottom, agarose gel of products. Primer pairs used are indicated above the gel. Predicted sizes of PCR fragments (in bp) are indicated under the gel and blue arrows point to the predicted migration position. In addition primers targeting exon 1 and 2 resulted in the presence of an additional band, longer by about 100 bp, consistent with the presence of an an alternative exonic sequence, situated between predicted exon 1 and exon 2.

atherosclerosis, thus warranting further investigation of the gene locus. Examination of public data indicated that the corresponding gene is predicted to span 42 kbp and encodes alternatively spliced transcripts characterized by some exon conservation but low coding potential, typical of a long non-coding RNA (S1 Fig). We confirmed the existence of annotated exons by qRT-PCR using pairs of primers spanning predicted intronic sequences in HEK293T cells, consistent with the presence of at least two distinct transcripts (referred to below as *RP11* for simplicity) (Fig 1B).

## Identification of a promoter region governing *RP11* expression

The multiple possible splice variants suggested a complex transcriptional output that may be important for function. In this regard, the ability of the dCas9 derivatives to modulate both transcript abundance and complexity via endogenous gene regulation, seemed appropriate for the study of *RP11*. First, this approach was contingent on the identification of the promoter region of *RP11*. Exploiting annotated chromatin marks, proximity, ChiP and DNAse sensitivity, a ~ 1.5 kb region that included the putative exon 1 of *RP11* was identified (Fig 2A). In several cell types, the region is predicted to interact with general transcription factors (USF2, USF1, FOS) and core transcription machinery (POLR2A). To examine its ability to operate as a promoter, the corresponding region was subcloned upstream of a luciferase reporter and introduced in HEK293T and HeLa cells. Presence of the putative promoter conferred robust luciferase activity, reaching levels on par with promoters derived from *SV40* and *MFGE8*, a gene flanking *RP11* whose transcript is abundant and widely expressed (GTex) in both HeLa cells and HEK293T cells (Fig 2B). Functional validation of the putative promoter was performed using the CRISPRa and CRISPRi systems. Constructs expressing dCas9 fused to either the VP160 activator or the KRAB repressor were introduced into HEK293T cells, in conjunction with sgRNA expressing plasmids to target CRISPRa/i to the *RP11* promoter region. Transfection of CRISPRa and CRISPRi plasmids resulted in either activation or repression, respectively, of *RP11* expression that reflected the predicted cognate sgRNA binding site: the impact was maximal in the middle of the region and attenuated on either side (Fig 3); interestingly the region of maximal activation was central to reported DNase I hypersensitivity and ChIP Seq region. DNA accessibility seemed insufficient for activation as regions located closer to the transcription start site did not support activation, reflecting the presence of other

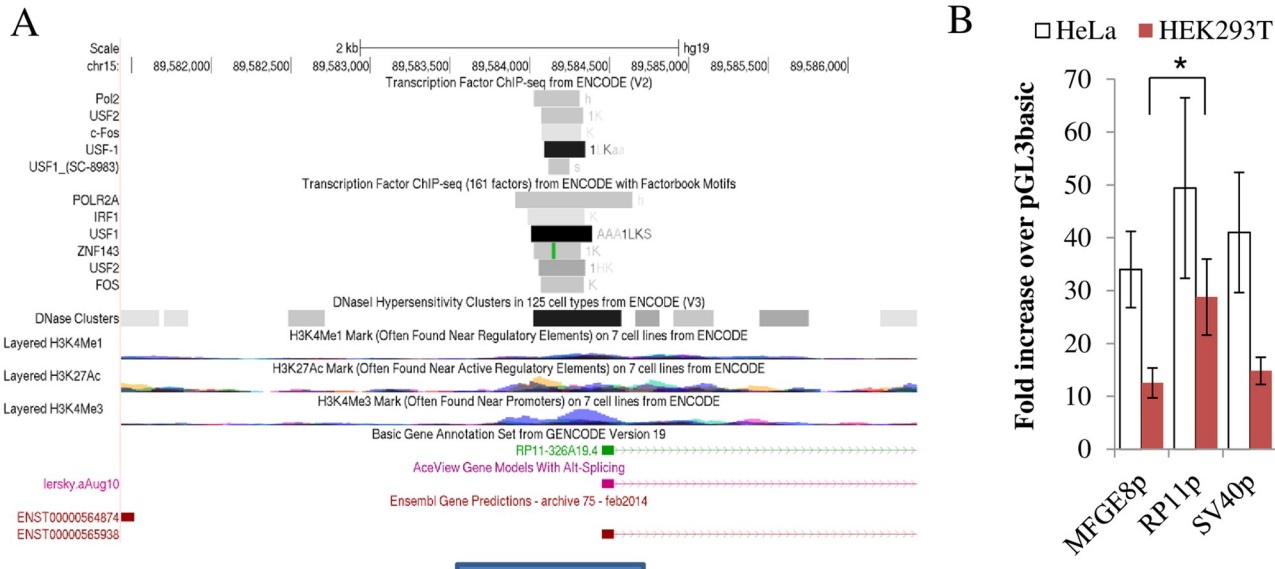

**Fig 2. Identification and characterization of putative *RP11* promoter.** A, Identification of putative promoter region. Promoter region (blue bar) used for reporter assay is shown under a 5 Kb UCSC browser snapshot of chIP-seq results, DNAse I hypersensitiviy and epigenetic marks from ENCODE. Putative promoter includes ChIPseq peaks as well as a DNAse rich region, as indicated. B, Proximal region upstream of *RP11* operates as a strong promoter in isolation. Activity of three promoter regions assessed by reporter assays, expressed relative to the promoterless control (pGL3basic). SV40, a strong promoter as well as the putative promoter region proximal to the *MFGE8* transcriptional start site are included for comparison. Data represent the average of 4–6 experiments for HeLa and 3 experiments for HEK293T (± 95% C.I.). All promoter constructs increased luciferase activity significantly (p = 0.05–0.001) relative to the pGL3 basic control. RP11p had levels significantly higher than MFGE8p (p <0.05) only in HEK239T.

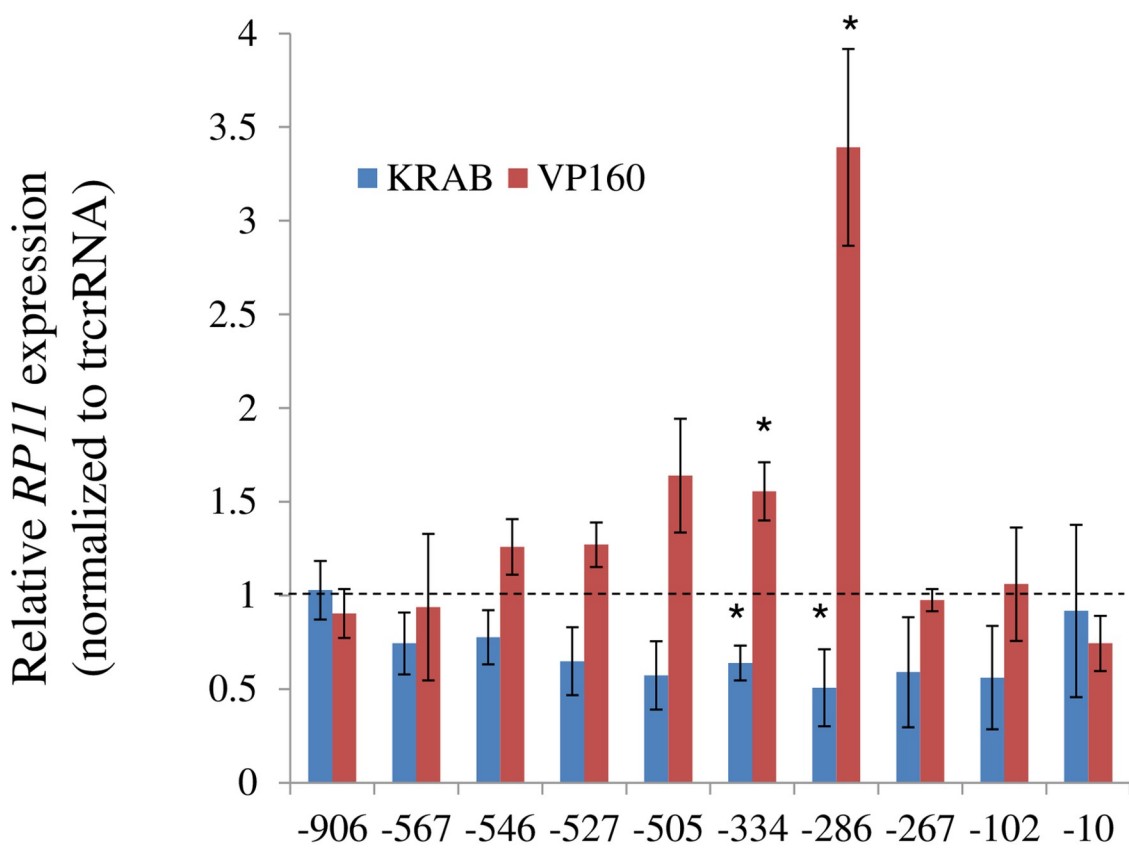

**Fig 3. Validation of the promoter region by CRISPRi and CRISPRa.** A, HEK293T cells were transfected with sgRNA expressing plasmids in conjunction with either Cas9 fused to a KRAB repressor (CRISPRi) or a VP160 activator (CRISPRa). Expression levels of *RP11* were measured 48 h later. Numbers on the x-axis refer to the predicted binding region of the sgRNA, relative to the transcription start site of *RP11-326A19.4*, set to 1. Expression values were normalized first to peptidylprolyl isomerase A *(PPIA)* values and then divided by the corresponding values from within experiment control transfected with trcrRNA and CRISPRa expressing plasmids. ChIPseq and DNAse I hypersensitivity sites are from ENCODE. Results represent the average of 3 biological replicates (± S.D.).

constraints (e.g. CRISPRa position relative to the transcription machinery, orientation, lack of binding etc. . .).

## Altered *RP11* transcription does not affect local gene expression

The impact on neighboring genes was investigated under the assumption that the *RP11* locus affects local transcription, in line with proposed models of lncRNA actions. As illustrated above (Fig 1A), the *RP11* locus is flanked by 2 protein coding genes *MFGE8* and *ABHD2* and further away by *HAPLN3*. The effect of *RP11* activation and repression on these as well as 2 control housekeeper genes, including *SRP14* located distantly on the same chromosome, was assessed by qPCR analysis. Transfection of SG$^{-286}$ along with CRISPRa or CRISPRi had no significant impact on either local or housekeeper gene levels (*SRP14* and *GAPDH*) (S2 Fig). Thus *RP11* levels and/or transcription activity do not appear to regulate regional transcriptional output.

## Genome-wide transcriptome interrogation of changes associated with CRISPRa of *RP11* reveals an upregulation bias

To identify putative *RP11* targets that might help elucidate its function, the transcriptome-wide impact of increased *RP11* transcription was assessed. We reasoned that effects secondary to *RP11* activation, be they at the transcriptional or post-transcriptional levels, should be mirrored by changes in the transcriptional fingerprint of the cell. Importantly, introduction of CRISPRa elicits a restrained *RP11* response which should minimize non-specific effect effects associated with typical transduction type experiments. Transcriptome profiles of HEK293T cells transfected with CRISPRa alongside SG$^{-286}$, which led to the strongest *RP11* increase (3–4 X), or a control tracrRNA construct, were compared using gene expression arrays (Clariom D). The analysis confirmed a 4.4 increase in *RP11*. Furthermore, focusing on nominally significant (p <0.05, > 2-fold linear change) and validated transcripts (no AceView transcripts), the analysis revealed 220 genes, of which only 39 were downregulated (S1 Table). The departure from a more evenly distributed repressive/stimulating profiles expected *a priori*, suggested that a large proportion, perhaps most, of the changes resulted from off-target binding and activation by CRISPRa. Alternatively, these results may point to a role of *RP11* in promoting transcription. Looking for possible genomic binding motif that could account for this, genome-wide interrogation of differentially expressed transcripts by Overrepresentation Enrichment Analysis (ORA) via WebGestalt indicated no FDR-significant enrichment motifs within 4 kb of the transcription start sites of impacted genes, although a few motifs reached nominal significance (Table 1). Thus changes in expression cannot be ascribed to the presence of a particular subset of transcription factor binding motifs within the genome.

## *IL6* transcript and protein levels are increased in response to CRISPRa targeting of *RP11*

Among the list of nominally affected genes (S1 Table), we noted the presence of 2 cytokines that hinted to a possible implication of *RP11* in the regulation of the immune response: *IL6* and *IL36B*. Both were confirmed to be upregulated by qPCR (Fig 4A for *IL6* (10.3-fold ± 3.2) and 3.7-fold (± 1.3) over control (n = 3, p = 0.07) for *IL36B*). As *IL36B* expression was very low in HEK293T (Cp ~ 36), we chose to focus on *IL6* whose transcript was readily detectable, albeit at low abundance by qRT-PCR (Cp~ 30). Comparable results were obtained with 2 non-

**Table 1. Over Representation analysis of transcription factor targets.**

| Motif | Size | Expect | Ratio | P Value | FDR |
|---|---|---|---|---|---|
| TGGNNNNNNKCCAR_UNKNOWN | 409 | 2.576 | 3.8821 | 0.000232 | 0.082281 |
| IK3_01 | 213 | 1.3415 | 5.218 | 0.000377 | 0.082281 |
| GTGGGTGK_UNKNOWN | 285 | 1.795 | 4.4569 | 0.000412 | 0.082281 |
| NKX61_01 | 226 | 1.4234 | 4.9179 | 0.000538 | 0.082281 |

Over Representation Analysis of transcription factor targets via WebGestalt. The analysis looks within a gene list for a particular pattern that is enriched in a subset of gene. In the current analysis, promoter sequences of the nominally affected genes and the entire gene list are first screened for the presence of putative as well as validated transcription factor binding sites. Binding sites that are enriched in the normally affected list (i.e. more present than expected by chance) may indicate shared underlying effectors. RNAs isolated from HEK293T cells transfected with dCas9-VP160 alongside either trcrRNA or SG$^{-286}$ were subjected to transcription array analyses. Transcripts showing more than 2-fold and nominally significant (p<0.05) changes were mapped to 135 unique Entrezgene IDs; more speculative genes (Pseudogenes, lncRNA and Aceview transcripts) were excluded from the analysis. Of the 135 IDs, 76 were annotated to the functional categories and were present in the reference list. The reference list was mapped to 21491 Entrezgene IDs of which 12067 were annotated to functional categories. Multiple comparison correction was performed by False Discovery Rate according to Benjamini Hochberg and only the FDR values lower than 0.1 are shown; FDR significance threshold is 5% (0.05). Size, number of genes in the reference list; Expect, expected representation; Ratio, ratio of Obtained/Expected; P value, nominal p value; FDR, False Discovery Rate (q value).

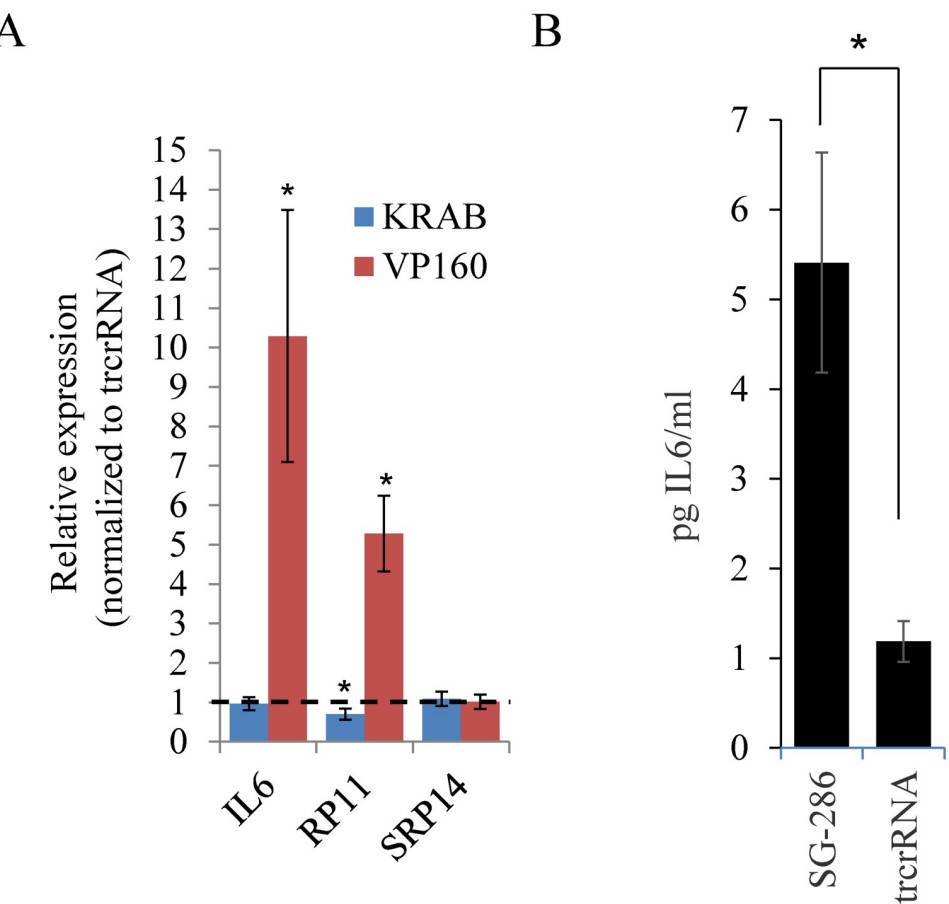

**Fig 4. CRISPRa affects IL6 expression in HEK239T.** A, regulation of *RP11* transcription by repressive (KRAB) or activating (VP160) Cas9 derivatives. Cells were transfected with an activator or a repressor, in the presence of SG$^{-286}$ or trcrRNA alone. Set used was independent from Fig 5. Data from 3 independent experiments (except for [*IL6/ RP11*] + KRAB, 5 experiments). B, CRISPRa activation increases secreted IL6 levels. IL6 protein concentration was measured in the media of HEK293T cells transfected for 48 h with CRISPRa and either SG$^{-286}$ or trcrRNA alone.

overlapping pair of *IL6* qPCR primers, ensuring detection specificity (See S1 Supplementary Materials for primer information). Moreover, increased transcript levels were accompanied by higher IL6 protein concentration in the media, as assessed by enzyme-linked immunosorbent assay (ELISA) (Fig 4B). In contrast to CRISPRa, CRISPRi (KRAB), which in combination with SG$^{-286}$ inhibited *RP11* expression by ~30%, had no noticeable impact on *IL6* (Fig 4A). The relatively small impact on *RP11* is likely due to the design of SG$^{-286}$, which is optimized for activation (upstream of the transcription start site) rather than inhibition (internal to the transcript). These results suggest that *RP11* upregulation, but likely not its downregulation, may regulate *IL6* expression.

### *IL6* is increased in response to CRISPRa using sgRNA designed against distinct regions of the *RP11* promoter

CRISPR/Cas9 related technologies suffer from specificity limitations and increased *IL6* associated with *RP11* upregulation could be the result of off-target CRISPRa binding. To more specifically link transcription activation of *RP11* by CRISPRa to *IL6*, levels of *IL6* were examined in response to multiple sequences covering the promoter region. Moreover, a distinct

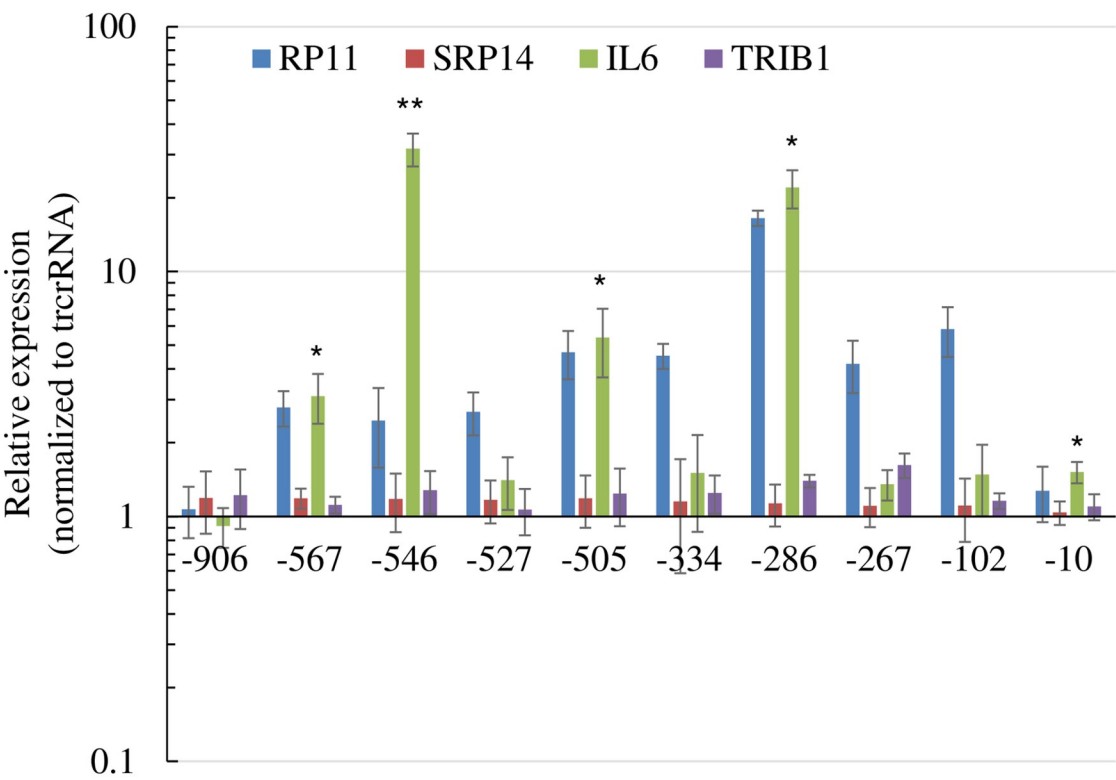

**Fig 5. Increased expression of *IL6* by CRISPRa directed toward several regions of *RP11* promoter.** HEK293T transfected with dCas9-VPR and sgRNA constructs were analyzed. Values for *RP11* represent the average from 3 distinct primer combinations spanning exons 1–2, 2–3, and 3–4. Two specificity controls (*SRP14* and *TRIB1*) are included. Expression values were normalized first to PPIA and then to a within experiment control transfected with dCas9-VPR and the empty trcrRNA expressing plasmid. Results represent the average of 3 biological replicates (± S.D.). For simplicity only statistically significant tests for *IL6* are shown (*: p<0.05; ** p<0.01).

activator (VPR vs VP160) was used, which resulted in more potent and pervasive activation of *RP11* (Fig 5). Significantly, increased *IL6* was observed with half of the sgRNAs. By comparison, two other control transcripts (*TRIB1* and *SRP14*) were only modestly affected, although some reached nominal statistical significance (*TRIB1*, SG$^{-286}$ and SG$^{-267}$).

## The *RP11* promoter region is rich in *IL6* activation sequences

Despite some inconsistencies (e.g. SG$^{-546}$), which were inferred to stem from off-target binding to the *IL6* gene, these experiments argued for a functional relationship between the *RP11* locus and *IL6*. Presumably, this could reflect a general propensity of *IL6* to respond to activation by CRISPRa. To examine this possibility, the *MFGE8* promoter situated 130 kb upstream of the *RP11* gene was in turn targeted by CRISPRa. We used an approach similar to that of the *RP11* locus, whereby multiple sgRNA sequences were used to capture any possible off-target effects. For a subset of sgRNA, targeting dCas9 to *MFGE8* resulted in increased *MFGE8* expression. However, increased transcript output was not accompanied by increased *IL6* levels (S3 Fig). This underscores the particular functional relationship of *IL6* with the *RP11* promoter region.

## *IL6* induction by CRISPRa does not require the *RP11* locus

To assess the role of *RP11* in controlling *IL6*, exon 1 of *RP11* along with its proximal regulatory region was deleted using CRISPR/Cas9 by simultaneously targeting the SG$^{-286}$ and a sequence

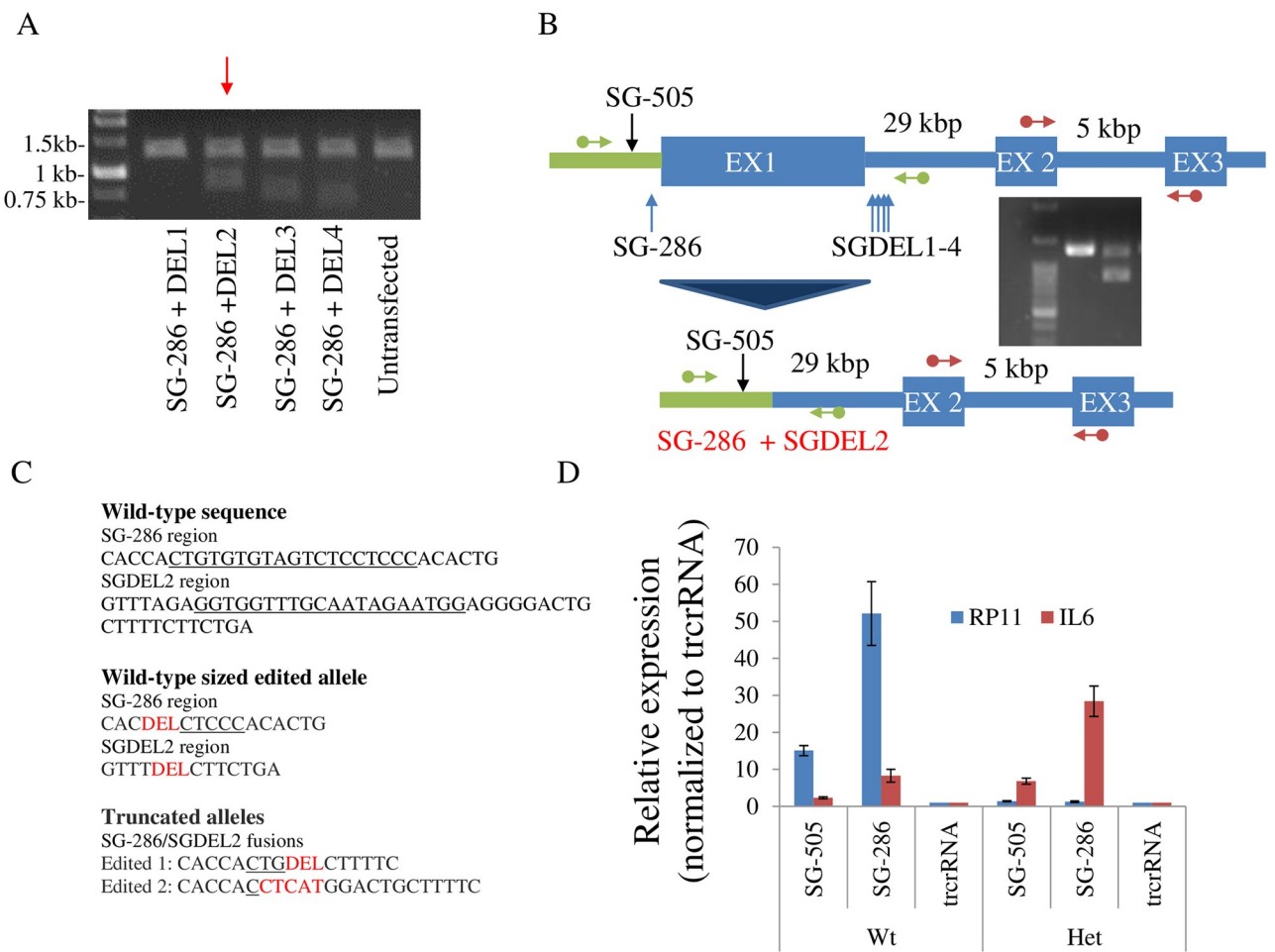

**Fig 6. CRISPR/Cas9 editing of *RP11* abrogates *RP11* induction but does not prevent *IL6* induction by CRISPRa.** A, removal of a region spanning 418 bp by CRISPR/Cas9. Expected size of native fragment is 1.3 kbp. Using 2 distinct sgRNA, deletion by Cas9 results in appearance of a novel band of reduced size (~ 0.9 kbp for SG$^{-286}$/DEL2 combination). B, schematic of the deletions. Green arrows indicate position of PCR primers used to validate gene editing. Red arrows indicate qRT-PCR primers (recognizing exon 2 and 3) used in D. Drawing not to scale. C, Sanger sequencing results of the edited alleles. For the Wt sized allele, a single deletion pattern was observed wherein both sgRNA recognition sites were deleted extensively. As for the shorter allele species, two distinct sequences, all with edited recognition sites, were observed. The cognate sgRNA sequence, or residual sequences thereof, are underlined. Deletions (DEL) and insertions are indicated in red. D, Impact of deletion on compound heterozygotes cells targeted by CRISPR/Cas9 as in A, using SG$^{DEL2}$ and SG$^{-286}$ sgRNAs. A mixture of 3 heterozygote clones were compared with 3 Wild-type clones (treated with CRISPR/Cas9 only) Data represent the average of 3 experiments, performed over 3 passages (± S.D.).

within intron 1. Unfortunately, the process yielded only compound heterozygotes ([Fig 6]). Sequencing of the Wt sized allele revealed that both sgRNA cognate sites employed for deletion were extensively edited. Unexpectedly, despite retention of an otherwise intact *RP11* allele, *RP11* became resistant to CRISPRa induction by SG$^{-505}$, whose binding is predicted to occur upstream of the deletion. Importantly, cells retained the ability to induce *IL6* in response to CRISPRa. These results indicated that the *RP11* transcript was dispensable for *IL6* induction, but left open the possibility that interchromosomal contacts could play a role. Alternatively, this could indicate that activation was through off-target binding by CRISPRa. Specificity was first tested for SG$^{-286}$, whose cognate sites were absent in the compound heterozygotes. Surprisingly introduction of dCas9-VPR alongside SG$^{-286}$ still augmented *IL6* levels, thus demonstrating off-target effects, at least for that sgRNA. In light of these findings, the specificity of

SG$^{-505}$ was tested by generating a SG$^{-505}$ binding site knock-out (using Cas9 and SG$^{-505}$), which again revealed that *RP11* but not *IL6* upregulation was abrogated by the deletion (S4 Fig). Together these findings indicate that *RP11* and *IL6* are functionally uncoupled and that impacts on the latter are mediated by off-target mechanisms.

### Off-target prediction algorithms fail to identify *IL6* as a direct target of CRISPRa

However surprising for 2 distinct sgRNA sequences, these observations suggested that increases in *IL6* and *RP11* share similar underlying processes, i.e. sgRNA steers dCas9-VPR to the *IL6* locus. Indeed we observed a slow accumulation of *IL6* which paralleled *RP11*'s and which contrasted with classical *IL6* induction mechanisms which involve acute responses (Fig 7). Searches using commonly used off-target algorithms trained on CRISPR sgRNAs failed to identify SG$^{-286}$ or SG$^{-505}$ sites matching *IL6* [19,20]. Unfortunately, these algorithms are optimized for prediction of relative (off-target) cleavage rather than binding. Similarly, tools incorporating CRISPRa predictions allow only for a few mismatches from the optimal 20 mer consensus and failed to identify any off-target [21,22]. CROP-IT which allows for more extensive misalignments and binding (up to 10 mispaired nucleotides) identifies ~110,000 genome-wide binding sites for SG$^{-505}$ and SG$^{-286}$ [23]. However, none were in the vicinity of *IL6* (S2 Table). If bulges in RNA and/or DNA are allowed [24], the number of binding sites increases

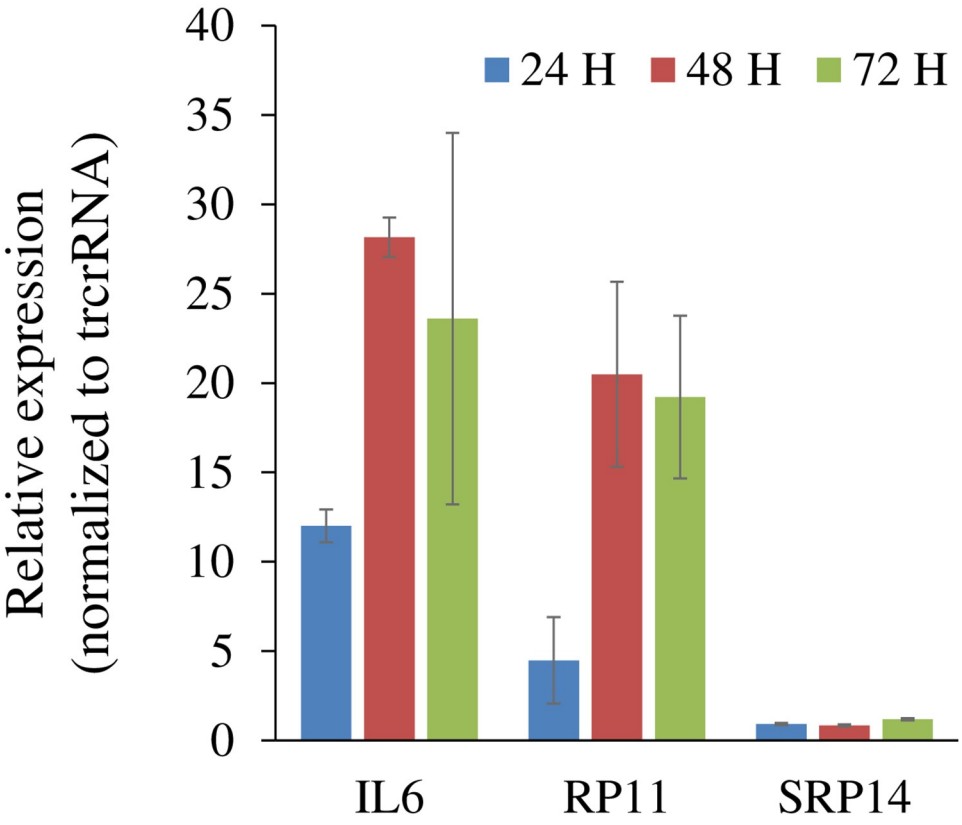

**Fig 7. Upregulations of *RP11* and *IL6* occur in parallel.** Time course of SG$^{-286}$ activation in HEK293T. Cells were transfected with dCas9-VPR and SG$^{-286}$ or trcrRNA for the indicated times. Indicated RNAs were quantified by qRT-, normalized to *PPIA* values and then divided by the trcrRNA/dCas9-VPR. Data represent the average of 3 experiments (± S.D.).

dramatically (S3 Table). For instance, allowing a 5 nucleotide mismatch and 1 RNA bulge on a 20-mer sequence allows for ~600,000 potential targets or 1 binding site every ~5.5 kb bases in a 3.3 Mb human diploid genome, or on average ~ 5 binding sites for a "typical" 28 kb gene [25]. Thus, current prediction tools are unfortunately of limited utility to recognize *in situ* off-target dCas9 binding sites.

## Activation of *IL6* by CRISPRa involves multiple regulatory pathways

Having demonstrated induction of *IL6* by this experimental approach, we turned our attention to the underlying regulation, anticipating that identification of mechanisms at play may shed light on possible upstream regulators and *IL6* regulation in general. Physiological *IL6* expression is controlled by several upstream kinases including MAP kinases, PI3K and IKK. The contribution of these regulators to CRISPRa-mediated induction was tested first by targeted inhibitions in HEK293T cells. Using two distinct sgRNAs, all drugs reduced *IL6* expression relative to the corresponding vehicle-treated CRISPRa, although the impact of TCPA-1 (IKK), Wortmannin (PI3K) and PD98059 (MAPKK) were more pronounced and reached statistical significance (Fig 8 and S5 Fig for *SRP14* control). Interestingly SB203580 (p38 MAPK inhibitor) affected induction by one sgRNA but not the other, hinting that the underlying mechanisms differ in part. By contrast *IL6* levels in the control CRISPRa/ trcrRNA samples remained unaffected by the treatments, indicating that these kinases play no role in supporting basal *IL6* expression. These findings appear consistent with the activation by CRISPRa involving established *IL6* regulators.

## CRISPRa does not induce measurable activation of *IL6* master regulators

Activation status of these pathways was tested next. As a proxy for activation, phosphorylation levels of ERK (downstream target of the MAPKK axis), IKK, AKT (a downstream target of PI3K) and p38 were examined in response to CRISPRa. Of note, these effectors typically exhibit rapid and transient kinetics that differ from our slower transfection based approach. Still, we reasoned, based on the inhibitor studies, that the steady accumulation of *IL6* resulting from the introduction of CRISPRa may reflect enduring activation of these effectors. Introduction of CRISPRa was accompanied by marginal phosphorylation changes, none of which were directionally consistent with the *IL6* activation profile ($SG^{-286} > SG^{-505} > SG^{-10}$) (Fig 9, S6 and S7 Figs). Thus, increased *IL6* is not accompanied by a measurable impact on its predicted regulators. Together with the inhibitor findings, these results point to the activation of established *IL6* regulatory pathways, albeit to an extent that is not detectable by our methods. Alternatively, the contribution of distinct effectors sharing sensitivities to the inhibitors cannot be ruled out.

## Regulation of *IL6* by CRISPRa in a physiological context

We hypothesized that CRISPRa, by artificially inducing *IL6* expression via a low grade response may interact, either positively or negatively, with the normal inflammatory response machinery and *IL6* responsiveness. *IL6* is a well-characterized target of NFKB, a prototypical inflammatory response transcription factor [15]. Upon activation, NFKB dissociates from its dedicated repressor, migrates to the nucleus and binds to specific response elements present in the promoters of a number of genes, including *IL6*. TNFα is a major proatherogenic regulator and a classic activator of NFKB via the ubiquitously expressed TNFR1receptor 1 (TNFR1) [26]. Thus the response of *IL6* to TNFα was examined in the context of CRISPRa activation, anticipating that CRISPRa might impact TNFα activation of *IL6*. Unfortunately, HEK293T cells were found unsuitable since *IL6* levels therein did not respond to TNFα treatment (2 h,

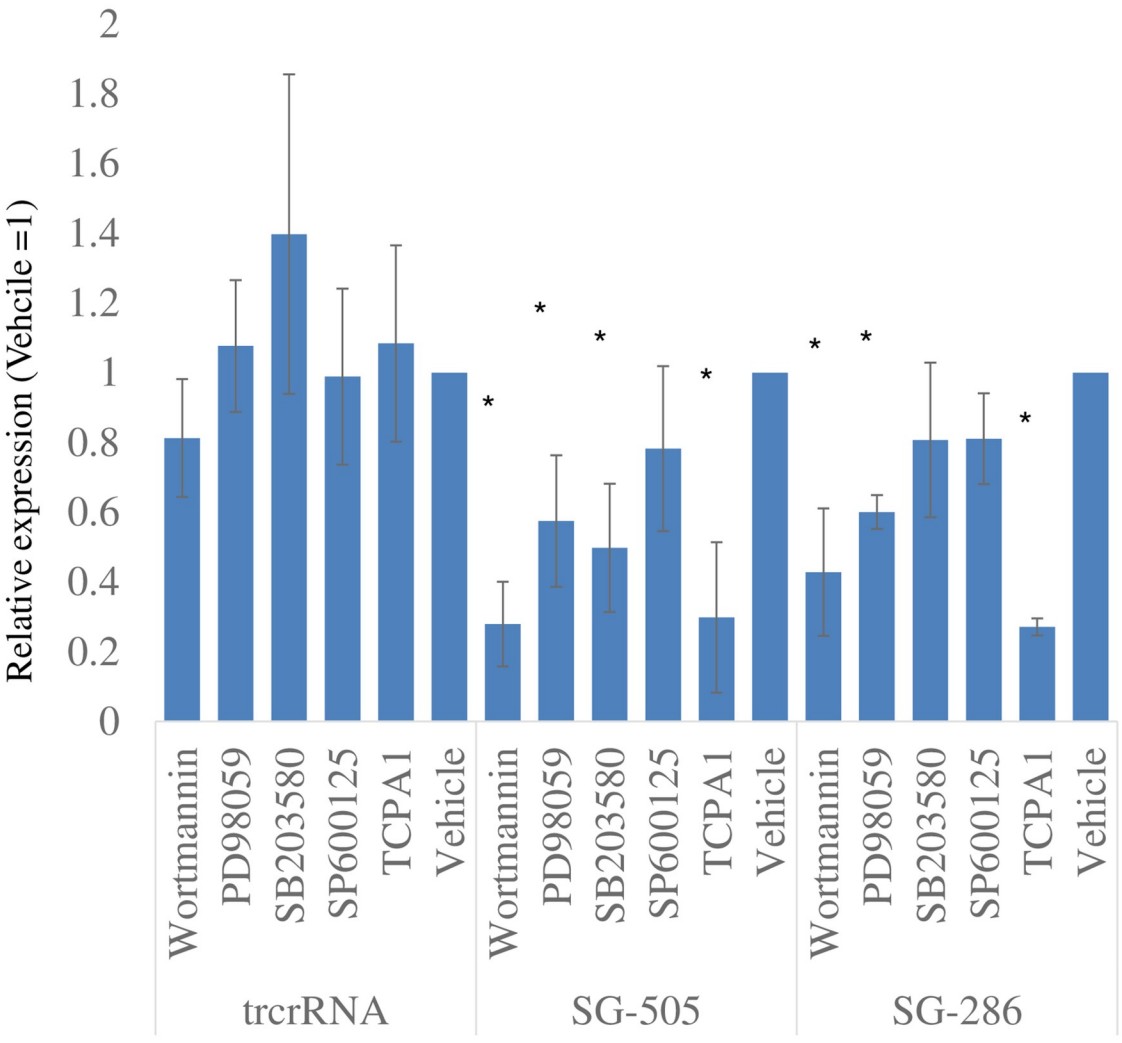

**Fig 8. Kinase inhibitors interfere with CRISPRa activation of IL6.** HEK293T cells were transfected for 24 h with dCas9-VPR together with the indicated sgRNA constructs, in the presence of kinase inhibitors. Cells were then harvested and *IL6*, *SRP14* and *PPIA* levels were quantified. Values for IL6 (relative to PPIA) are shown; SRP14 quantifications are shown in S4 Fig. Values are normalized to the corresponding vehicle values to emphasize the impact of the drugs on each CRISPRa transfection and facilitate inter-construct comparisons. Data represent the average of 3 experiments (± S.D.).

30 ng/ml; 1.2-fold (± 0.24) over vehicle (n = 2). Rather, the impact of CRISPRa on *RP11* and *IL6* was examined in a different cell type, HuH-7, a hepatoma cell line widely used to study hepatocyte function. Inflammatory components, including *IL6*, have been shown to play crucial roles in normal and pathological liver functions [27]. As in HEK293T, SG$^{-286}$ and SG$^{-505}$ increased *RP11*, although the fold-increase was more dramatic, possibly due to lower basal expression of *RP11* in those cells (S8 Fig). HuH-7 cells were also responsive to TNFα, at least acutely: addition of TNFα for 2 h resulted in increased *IL-6* in naïve or mock transfected cells, although statistical significance was only reached for mock (Lipof3000) transfected cells (Fig 10). By contrast, longer treatments had no noticeable impact on *IL6*. Introduction of dCas9-VPR/SG$^{-286}$ was sufficient to increase *IL6* to levels on par with those reached with TNFα treatment alone. Combining dCas9-VPR/SG$^{-286}$ with TNFα conferred an additional increase, albeit without noticeable synergy. These results clearly demonstrate that CRISPRa

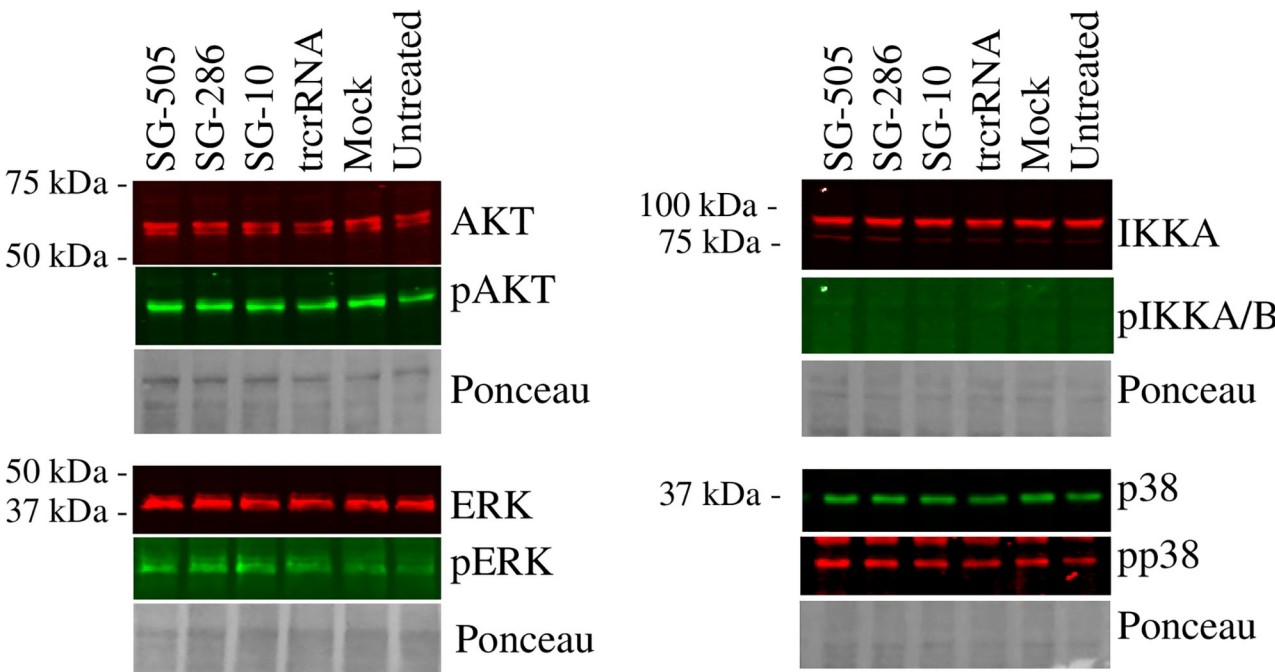

**Fig 9. Activation by CRISPRa occurs in the absence of detectable changes in known *IL6* regulators.** HEK293T cells were transfected with dCas9-VPR and the indicated SG constructs for 24 h, lysed and analyzed by Western blotting. Quantification is presented in S6 Fig. PhosphoIKKA/B signal was too low to be confidently identified and quantified; a positive control for phosphoIKKA/B (M1 polarization of THP-1 cells) is shown in S7 Fig.

does not interfere with normal TNFα signaling, insofar as *IL6* is concerned, but rather suggest that both contribute, independently, to *IL6* upregulation.

## Discussion

In a landmark study, systematic dissection of 12 lncRNA loci demonstrated that for all loci investigated, transcriptional activity *per se* rather than nature of the transcript affects neighboring protein coding genes [28]. CRISPRa/i approaches offer the ability to regulate *in situ* transcription with unparalleled ease and thus facilitate research on lncRNA. Using CRISPRa/i, we demonstrated that expression of neighboring genes was resistant to *RP11-326A19.4* perturbation. Rather a long distance impact on several transcripts, including *IL6*, was observed. We chose to focus on *IL6*, based on its known importance in human coronary artery disease. Importantly, validation studies indicated that increased *IL6* was the result of off-target activity by CRISPRa.

The observation that 4 out of 10 sgRNA targeting *RP11* lead to increased *IL6* is intriguing and could be construed to indicate specificity. To our surprise we found that when tested via binding site abrogation, 2 different sgRNA were non-specific, i.e. altered *IL6* via other means. The mechanism(s) remain unclear. In contrast to *IL6*, the same sgRNAs had little or no effect on controls (*SRP14* and *TRIBI*). To control for the promoter region of *RP11*, 10 other sequences designed to target the *MFGE8* promoter region had no or very modest effects on *IL6*. One possible explanation may be that the *RP11* promoter region is rich in sequences that are linked to *IL6* upregulation. Although sequence homology searches (BLAST) revealed no salient homology between the *IL6* gene and sgRNA used, the proclivity of dCas9 to bind promiscuously leaves open the possibility that it binds to *IL6* regulatory regions. In view of the

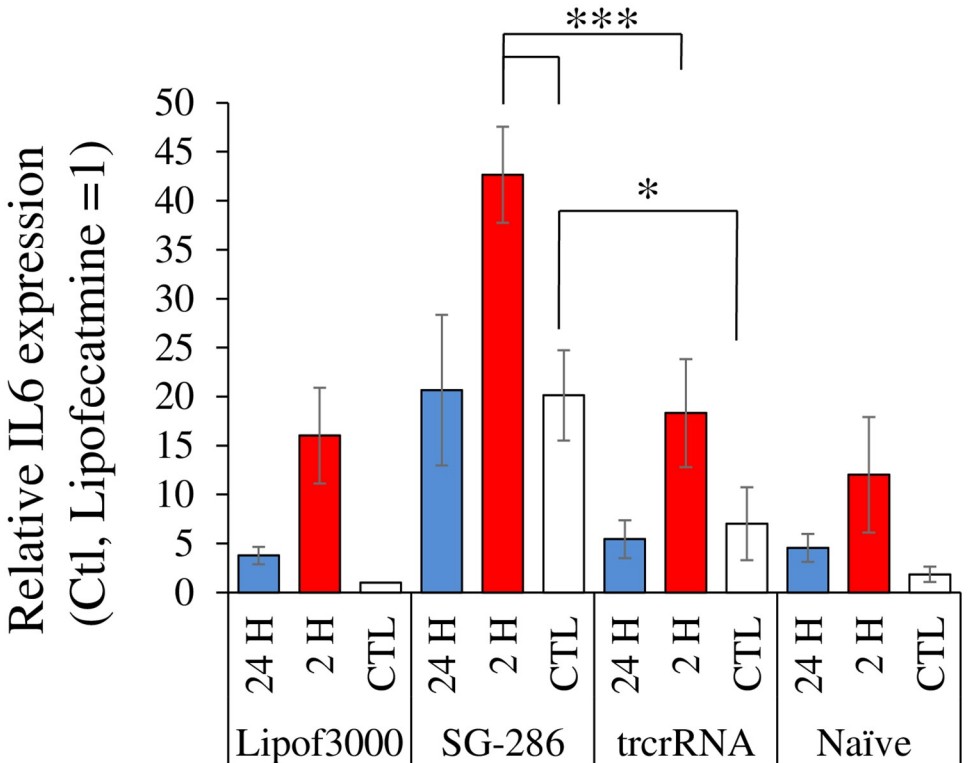

**Fig 10. Distinct effects of TNFα and CRISPRa on the stimulation of _IL6_ expression.** HuH-7 cells were transfected with dCas9-VPR and the indicated SG constructs for 48 h. TNFα (30 ng/ml) was included for the last 24 h (blue bars) or 2 h (red bars) of transfection, as indicated. Levels of _IL6_ and _PPIA_ were then measured by qRT-PCR. Pertinent statistical test (1-way ANOVA) results are shown (*, p <0.05; ***, p<0.001). Data represent the average of 3 biological replicates.

current bioinformatic limitations, teasing out CRISPRa target(s) mediating _IL6_ upregulation will require genome-wide interrogation of _in situ_ binding sites, followed by their functional validations.

While the events surrounding CRISPRa binding are unclear, our study demonstrates that _IL6_ upregulation involves traditional _IL6_ master regulators. This conclusion is based on inhibitor evidence, which blocked _IL6_ upregulation. However, as phosphorylation remained below detection, increased _IL6_ could be not be linked to the activation status of the regulators. This could simply reflect the transient nature of the activation which might be reversed at the time of analysis. However we note that activation continues beyond this time point, which seems inconsistent with that interpretation. The progressive, long-term accumulation of _IL6_ versus the more conventional transient response typically studied, points to an alternative explanation. We speculate that these regulators are activated minimally, yet sufficiently to induce a slow and continuous rise in _IL6_. This accumulation of _IL6_ most likely involves both increased output as well as stabilization at the mRNA, both of which are known to control _IL6_ and immunokine levels in general [29].

dCas9 derivatives are increasingly being leveraged to interrogate biological mechanisms. For instance to understand and manipulate enhancer function, complementary dCas9 variants can be targeted to different regions of the genome thereby forcing their juxtaposition [30]. Correct interpretation of the generated datasets relies on unmitigated binding localization which, as shown here, this technology does not allow. More significantly, therapeutic

interventions involving activation of transcription suppressed genes, great care should be exercised given the significant possibility of off-target interactions.

The off-target effects reported here are for *Staphylococcus pyogenes* Cas9 derivatives, arguably the most commonly used form, largely for historical reasons. Other Cas variants have different cutting specificities but their applicability to multiplexed applications, where binding rather than cutting is the parameter of interest, remains to be as thoroughly investigated as spCas9. In order to obtain robust and truly useful tools, the dCas methodologies must be further improved. Indeed as little as a 5 mer + PAM sequence might be sufficient to confer binding to d(sp)Cas9 in mouse embryonic stem cells [5]. This will require binding optimization, including greater specificity and affinity. The use of shorter oligonucleotides was shown to improve cleavage specificity somewhat but the suitability of this approach to minimize off-target binding is doubtful [31]. Wild-type enzymes have evolved for both binding and cutting, and considerable room exists for molecular engineering approaches to favor binding specificity, likely at the expense of cutting. For instance, Cas9 derived from *Neisseria meningitis* is a recently described compact Cas9 that exhibits higher cutting fidelity and would be of potential interest in this regard [32]. By contrast, Cas9 from *Staphylococcus aureus* has been recently shown to be undergo faster turnover which may translate into weaker substrate affinity and mitigate its ability to be effective as CRISPRa/i [33].

This study includes an early investigation of *RP11-326A19.4* function which future work will aim to enrich by identifying *bona fide* effectors. Activating the *RP11* locus via CRISPRa/SG$^{-286}$ led to an upregulation bias, hinting that a substantial subset of the affected genes may in fact stem from off-target binding. To aid with on-target identification, expression changes induced by other *RP11* targeting sgRNA will be compared and contrasted [11]. The observation that *IL6* was independently upregulated with two sgRNA targeting *RP11*, reinforces the need for multiple additional sgRNA. Lastly, orthogonal interventions such as antisense oligonucleotides or *RP11* transduction are currently underway to examine the role of the transcript proper.

## Materials and methods

### Tissue culture and drug treatments

Cells were maintained in Dulbecco's Modified Eagle Medium (DMEM) containing either 4.5 g/l glucose (HEK293T, HeLa) or 1 g/l (HuH-7). HEK293T and HeLa were obtained from the ATCC and HuH-7 were purchased from Japanese Collection of Research Bioresources Cell Bank (JCRB Cell Bank). Unless mentioned otherwise all transfections were performed for 48 h using Lipofectamine 3000 (Thermofisher) at ratios of 3:2:1 (Lipof 3000: P3000: DNA). For inhibitor studies, the following drugs were used (working concentration): PD98059 (20 μM), SB203580 (5 μM), SP600125 (1 μM) and Wortmannin (1 μM) were from Cell Signaling Technology. TPCA-1 (20 μM) was from Sigma. Transfections were performed for 24 h in the presence of the inhibitors or vehicle (0.1% DMSO); inhibitors were added 30 min prior to transfection.

### Transcription array analysis

Transcriptome profiling on SG$^{-286}$/CRISPRa transfected cells was performed at the Centre for Applied Genomics (The Hospital for Sick Children, ON, Canada). Purified RNA, was converted to cRNAs and hybridized to Human Clariom D arrays. Results were analyzed by using the Expression and Transcriptome Analysis Consoles (Applied Biosystems); the underlying expression data set is accessible as GSE132451 at the Gene Expression Omnibus (GEO) repository.

## Luciferase assays

HEK-293T were seeded in 24 well plates and transfected 48 h later with 0.5 µg DNA using Lipofectamine 3000; pRSV40 (2% of total plasmid) was included as internal control. Cells were transfected for 24 h, lysed and processed for luminescence analysis, measuring firefly and renilla signals in a GLOMAX plate reader.

## ELISA

Media from HEK293T cells transfected with for 48 hours with CRISPRa (VPR) alongside either SG$^{-286}$ or trcrRNA constructs, were recovered. Following a 2 min centrifugation at 2000 x g, the cleared supernatants were frozen at -20 C until further analysis. Media were then thawed and assayed for IL6 using a commercial ELISA kit (HS600C, RandD systems) according to the supplier's protocol.

## CRISPRa and CRISPRi

Two activators were used, employing distinct humanized dCas9 versions (Addgene plamids 63798 and 48240). VP160 consists of 10 repeats of the p16 herpes virus activator while VPR is a fusion of p16, human herpes virus 4 (Epstein-Barr virus) replication and transcription activator Rta/BRLF and the transactivation domain of the p65 subunit of NFKB. While full-length NFKB is a known activator of *IL6*, VPR contains only its transactivation domain and has thus lost its ability to interact with cognate DNA sequences and effect transcription of its native targets. For CRISPRi, dCas9 fused to the Krueppel-Associated Box (KRAB) domain was transferred from Addgene plasmid 46911 into PLVX-puro (Clontech); for consistency, all constructs were introduced by transfection. Design of sgRNA for CRISPRa/i was performed via the Sequence Scan for CRISPR (SSC), using a 19 nt spacer sequence default and CRISPR activation criteria.

## Western blot analyses

Whole cell extracts were lysed in SDS-free RIPA buffer (50 mM Tris-HCl, 0.15 M NaCl, 1% Triton X-100, 0.5% NaDeoxycholate, pH 7.4) supplemented with 1 X Complete Protease Inhibitor Tablets and 1 X phoSTOP cocktail (Roche). Lysates (~30 µg) were cleared by centrifugation for 2 min at 4 ˚C, denatured in reducing SDS-PAGE sample buffer and resolved on 1 mm thick 4–15% gradient SDS-PAGE gels; protein ladder was from Bio-Rad (Precision plus All blue). Proteins were then transferred to nitrocellulose membranes for 40 min at 100 Volts in transfer buffer containing 15% ethanol. Blocking was performed for 1 h in PBS blocking buffer (Li-Cor) diluted 1:1 with ddH20, followed by incubations with primary antibodies (1:2000) overnight. Incubations with secondary antibodies (Li-Cor; 1:20,000) were for 1 h. All antibodies were diluted in PBS and washes were performed in PBS/0.1% Tween-20. Antibodies are detailed in S1 Supplementary Materials. Acquisition and quantification was performed on an Odyssey imaging station equipped with its application software version 3.0.21.

## CRISPR targeting of *RP11*

Deletion of exon 1 in HEK293T was accomplished by CRISPR using sgRNA sequences located within the promoter of *RP11* and intron 1. Editing at the SG$^{-505}$ were obtained via a similar approach, albeit using SG$^{-505}$. Briefly, cells were transfected with Addgene # 71814 (eSpCas9 (1.1)) alongside sgRNA expression plasmids at a mass ratio of 7/3 (Cas9/sgRNA). Single clones were then obtained by serial dilution and expanded. Editing was validated by the Guide-it Mutation detection kit (Takara Bio) and by the disappearance of a PCR product targeting the intact sgRNA site. For exon 1 targeting, 48 clones were obtained of which about half showed

some evidence of editing over the targeted area; only three clones had deletions consistent with the loss of one promoter region allele. No clones harboring bi-allelic deletions of the promoter region were obtained. For SG[-505] editing, 12 clones were screened, and 3 showed editing by Guide-it and 1 clone had bi-allelic editing. Editing was confirmed by sequencing of PCR amplicons obtained using oligonucleotides flanking the predicted deletion, after transfer into pCR2.1-TOPO (Thermofisher). Oligonucleotide sequences used to generate sgRNA are detailed in S1 Supplementary Materials.

## RNA isolation and qRT-PCR

RNA was isolated and converted to cDNA using the High Pure Isolation Kit (Roche) and the Transcriptor First Strand cDNA Synthesis Kit (Roche), respectively. For cDNA synthesis, a 1:1 mixture of random hexamer and oligodT was used. PCR amplification and quantification was performed using the SYBR Green I Master reaction mix (Roche) on a Roche LightCycler 480. Relative amounts of target cDNAs were normalized to peptidylprolyl isomerase A (*PPIA*). Oligonucleotides used are described in S1 Supplementary Materials.

## Statistical analysis

Unless mentioned Student's 2-tailed t-tests were performed to test for statistical significance. Results were deemed statistically significant if $p < 0.05$.

## Online resources

Integrated DNA Technologies CRISPR-Cas9 guide RNA design: https://www.idtdna.com/site/order/designtool/index/CRISPR_PREDESIGN
 SSC: http://cistrome.org/SSC/
 CRISPRscan: https://www.crisprscan.org/
 CRISPR-ERA: http://crispr-era.stanford.edu/
 CCtop: https://crispr.cos.uni-heidelberg.de/
 The Genetic Perturbation Platform Web Portal: https://portals.broadinstitute.org/gpp/public/analysis-tools/sgrna-design
 Cas-OFFinder: http://www.rgenome.net/cas-offinder/
 CROP-IT: http://www.adlilab.org/CROP-IT/cas9tool.html
 UCSC genome browser: https://genome.ucsc.edu/index.html
 WebGestalt: http://www.webgestalt.org/.

## Supporting information

**S1 Fig. Conservation of *RP11-326A19.4*.** UCSC browser snapshot of conserved regions spanning RP11-326A19.4. Conservation over *RP11* exons is restricted to mammals but varies with clades and species. For instance exon 1 is not found in most rodents (Euarchontoglires) but found in more distantly related Afrotheria. Exon 4 shows the highest conservation. The alternate TCONS forms (Human Body Map) reported exhibit less conservation in their last exon (red).
(PPTX)

**S2 Fig. No evidence of *cis* regulation by *RP11* in HEK293T cells.** Expression levels of genes proximal to *RP11* were assessed after *RP11* levels were modulated with either VP160 or KRAB in the presence of SG[-286]. Results are from 4 independent biological replicate (Average ± S.D.).
(PPTX)

**S3 Fig. *MFGE8* activation does not correlate with *IL6* expression.** CRISPRa targeting the *MFGE8* promoter region in HEK293T cells. Numbers refer to position of sgRNA cognate site relative to start site (RefSeq: NM_001114614). Data represent the average of 3 experiments (± S.D.).
(PPTX)

**S4 Fig. CRISPR targeting of the cognate SG^-505^ binding site within the *RP11* promoter prevents *RP11* but not *IL6* upregulation by CRISPRa.** Top, CRISPR editing of the SG-505 recognition region. DNA from a putative knock-out clone was PCR amplified, subcloned and individual clones sent for validation by Sanger sequencing. A total of 6 distinct sequences from that clone were obtained, conforming to 2 distinct patterns indicated above. One allele carried point mutations (red) near (-8 to -6) the PAM site while the other allele exhibited a more extensive deletion (DEL). Resulting sequences are expected to be poorly or not recognized by SG-505. Bottom, cells harboring bi-allelic editing of SG^-505^ binding site were transfected with CRISPRa (VPR) in the presence of either SG^-505^ or trcrRNA and levels of *RP11* and *IL6* (and *PPIA*) were measured 48 h post-transfection. Values are expressed relative to *PPIA* and then normalized to the corresponding trcrRNA values. Data represent the average values obtained from 3 passages.
(PPTX)

**S5 Fig. Kinase inhibitors do not affect SRP14.** Specificity control for Fig 8; see Fig 8 for additional information. Levels of SRP14 were measured in CRISPRa transfected cells (in the presence of SG-505, SG-286 or trcrRNA) and normalized to the matching vehicle (0.1% DMSO) CRISPRa sample. Data represent the average of 3 experiments (± S.D.).
(PPTX)

**S6 Fig. Impact of CRISPRa on known IL6 regulators.** Quantification of Western blots (Fig 9). Relative phosphorylation levels (pX/X) were measured and are normalized to the corresponding trcrRNA values (set to 1). Results represent the average of 3 biological replicates (± S.D.). *, statistically different (p < 0.05) from trcrRNA.
(PPTX)

**S7 Fig. Western blot positive control for pIKKA/B.** THP-1 cells, differentiated to M1 with phorbolesters (100 nM PMA, 72 h), were polarized to M1 with LPS (500 ng/ml) or vehicle (PBS, Ctl) for 2 h, and analyzed by Western blot for the presence of phosphorylated IKKA/B and IKKA. Arrowhead indicates predicted migration position of IKKA/B (84 and 87 kDa, respectively). Approximately 20 μg of THP-1 were loaded per well. HEK293T cells (30 μg, untreated) are included.
(PPTX)

**S8 Fig. Upregulation of *IL6* by *RP11* in a liver model.** Upregulation of *RP11* in HuH-7 correlates with increased IL6. HuH-7 cells were transfected with the indicated sgRNA expressing constructs, along with dCAS9-VPR. Values are expressed relative to trcrRNA values, set to 1. * indicates statistically significant (p < 0.05) change with Ctl, as assessed by ANOVA. Data represent the average of 3 experiments (± S.D.).
(PPTX)

**S1 Table. List of genes whose expression is nominally altered by CRISPRa/SG^-286^.** Complete searchable array results are available at the GEO repository (https://www.ncbi.nlm.nih.gov/geo/query/acc.cgi?acc=GSE132451).
(XLSX)

**S2 Table. CROP-IT results for SG$^{-286}$ and SG$^{-505}$.** Complete list of off-target sites predicted by CROP-it for both sgRNAs used (Tab 1:-286, Tab 2: -505).
(XLSX)

**S3 Table. Number of off-targets for SG$^{-286}$ taking bulges and mismatches into consideration.** Off-target predictions were performed using Cas-OFFinder.
(XLSX)

**S1 Supplementary Materials. Description of oligonucleotides and antibodies used in this work.**
(DOCX)

## Author Contributions

**Conceptualization:** Sébastien Soubeyrand, Ruth McPherson.

**Data curation:** Sébastien Soubeyrand.

**Formal analysis:** Sébastien Soubeyrand.

**Funding acquisition:** Sébastien Soubeyrand, Ruth McPherson.

**Investigation:** Sébastien Soubeyrand, Victoria Peters.

**Methodology:** Sébastien Soubeyrand.

**Project administration:** Paulina Lau, Ruth McPherson.

**Resources:** Paulina Lau, Ruth McPherson.

**Supervision:** Sébastien Soubeyrand, Paulina Lau, Ruth McPherson.

**Writing – original draft:** Sébastien Soubeyrand.

**Writing – review & editing:** Sébastien Soubeyrand, Ruth McPherson.

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
