## [Decision Letter · Decision Letter 0]

12 Aug 2019

PONE-D-19-18163

Off-target effects of CRISPRa on interleukin-6 expression

PLOS ONE

Dear Dr. Soubeyrand,

Thank you for submitting your manuscript to PLOS ONE. After careful consideration, we feel that it has merit but does not fully meet PLOS ONE’s publication criteria as it currently stands. Therefore, we invite you to submit a revised version of the manuscript that addresses the points raised during the review process.

We would appreciate receiving your revised manuscript by Sep 26 2019 11:59PM. To enhance the reproducibility of your results, we recommend that if applicable you deposit your laboratory protocols in protocols.io, where a protocol can be assigned its own identifier (DOI) such that it can be cited independently in the future. For instructions see: http://journals.plos.org/plosone/s/submission-guidelines#loc-laboratory-protocols

We look forward to receiving your revised manuscript.

Kind regards,

Serif Senturk, Phd

Academic Editor

PLOS ONE

Journal requirements;

1. We would recommend that you include in the methods section the source of all the cell lines used in your study.

Reviewers' comments:

Reviewer's Responses to Questions

5. Review Comments to the Author

Reviewer #1: Soubeyrand et al describe the finding of a CRISPRa off-target effect leading to increased IL6 expression while targeting a long non-coding RNA.

Major comments/suggestions

Line 99: There should be some explanation in the results section of the work leading to Figure 1.

Line 107: The authors should annotate the primers in the schematic representation of the locus. Furthermore, the authors need to explain how the results support the presence of an alternatively spliced exon.

Line 118: The authors should explain how the reporter construct is employed, and what it is.

Figure 3 Legend: The authors should explain the normalization strategy more clearly. I cannot understand what they mean by “Results are displayed as a function of the predicted binding region of the cgRNA, relative to exon 1 of RP11, set to 1”.

Line 127: If the region of maximal activation overlaps with the DNAseI hypersensitivity and ChIP Seq region, why is guide -267 and -102 not more active?

Line 149: Describe the experiment “Array analysis”

Line 212: The authors state that only heterozygotes were found. How many clones were tested? Is there an explanation why there were only heterozygotes? Where the clones heterozygote for the ~400pb deletion? Also, show the editing profiles.

Line 246: The authors should perform an unbiased off-target detection experiment, such as GUIDE-seq. This is not, as the authors state in line 323, a colossal undertaking. Furthermore, this can shed more light on the mechanism underlying the activation of IL6. I understand that multiple guides gave a similar phenotypic result (IL6 upregulation), but this could be explained by similarity in the promotor or enhancer regions.

Figure 7A: How was the normalization performed?

Minor comments/suggestions

Line 50: S.p.Cas9 does not require a single guideRNA (sgRNA) as it can also employ a two-part guide RNA consisting of a crRNA and a tracrRNA.

Line 54: What do the authors mean by suboptimal cleavage? Off-target sites can lead to high editing levels even in the presence of mismatches.

Line 57: Maybe the authors can refer to the dCas9 that is being used in this study.

Line 60: What do the authors mean by “...whose efficiency reflects its nuclease ability…”

Figure S1: Explain CARMAT.

Figure 2 Legend: The DNAse rich region is defined as a black bar. However, there are 3 different black bars in the figure.

Figure 3 Legend: Explain the following terms: PPIA & cgRNA.

Line 177: Which list are you referring to as “nominally affected”?

Line 189: The section head suggests that the RP11 promotor drives IL6 activation. This seems a bit strong.

Line 209: Should “CRISPR/Cas9” be replaced by “CRISPRa”?

Line 276: Can the authors describe the rationale of the experiments described in this paragraph?

Reviewer #2: This is a manuscript about the possible off-targeting problems relative to the usage of the emerging CRISPR activation system. The authors described the possible interference of the CRISPRa with the IL-6 cytokine, that can generate some concerns for the usage of this tool in future therapies. Although the idea is interesting, in my opinion the study needs some adjustments and clarifications before publication:

1) Why did the authors choose RP11 and not TCONS that are in the same Risky area?

2) Please, redesign Figure 1B because the scheme representing the primers binding is not clear and the results are not easy to read, make it clear. The PCR bands are too smeared.

3) Figure 2B: statistics analysis is missing. Please, explain why the MFGE8 promoter was chosen as control in this experiment.

4) Figure 3: y-axis the relative expression level is normalized to cells transfected with trcrRNA? This is not clear indeed in Figure 4A the y-axis name changes in “normalized to trcrRNA plasmid” that is not correct, because is it not possible to compare cDNA quantification with a plasmid. Moreover, it should be normalized using a housekeeping gene for the specific cell type.

5) The paragraph from line 163 to 174 is not clear please try to explain in a different way.

6) In the text, lines 184-185 the authors stated that after CRISPRi (using dCas9 fused to KRAB inhibitor) the RP11 expression was 30% lower with no effects on IL-6. It seems strange because it is known (Gao, X., Tsang, J. C., Gaba, F., Wu, D., Lu, L., & Liu, P. (2014) Nucleic acids research, 42(20), e155-e155.) that CRISPRi is even more effective for endogenous genes, and the off-target effect is mainly due to sgRNA used. This result is not demonstrating an off-target of the system because the RP11 can be an enhancer of the IL-6 gene (already demonstrated for lncRNA) and repressing it doesn’t mean that the endogen IL-6 is not transcribed.

7) Figure 4B: the authors showed that using different sgRNAs on RP11 promoter is leading to an increase of IL-6 due to CRISPRa. This seems real if consider only sgRNA -567, -505 and -286 (the guide with the strongest effect) but my concerned are about sgRNA -546, indeed using that guide the increase of RP11 seem to be not significant and the associated IL-6 is very high, can the authors speculate a little on this result? Moreover, if the off-target effect is not driven by sgRNA as explained with this figure, how can the deactivated Cas9 binds the DNA to drive the transcriptional machinery?

8) Figure 5: the authors tried to generate a partial promoter knock-out part of the 5’ UTR and exon 1, and they failed, “Unfortunately, the process yielded only compound heterozygotes”. My concerns regarding these experiments are due to the deletions: in three out of four deletions is resulted in heterozygotes (also if is not so clear, the smaller band seem two and not only one) and in the first one failed. Which cells did the authors use to perform the deletions? Maybe the problem is due to the presence of more than two copies of the genomic asset of the cell line? That should be clarified because the effect resulted can be also influenced by that. Moreover, I will suggest adding a chromatogram showing the junctions of the deletion, in fact using two sgRNAs is leading to a non-homologous recombination that can also interfere with the sgRNA -505 binding site. In Figure 5C the relative expression of RP11 and IL-6 in the wild type are opposed to what the authors showed in figure 4B, in Figure 5C the levels of RP11 in wild type are higher than IL-6 after transfection with both sgRNA-286 and -505. It diminishes the consistence of the data on heterozygous, thus the increment of IL-6 seem to be related to cells external or internal conditions.

9) In the Figure 6 is not clear why the authors did choose the SRP14 as a control. Moreover, they changed the relative for the normalization from transfected with the trcrRNA to PPIA. It would be more linear If they normalize everything to a fixed housekeeping. In addition, the standard deviation of the RP11 values at 72h is very high, maybe the experiment should be confirmed.

10) Did the authors try the alignment of the IL-6 and the RP11 promoters? I will advise doing it to understand the dissimilarity between them.

11) In Figure 7A, the authors want to demonstrate the inhibition effect of IL-6 transcription marking different kinases? These experiments are not clear and need to be represented in a better way. Moreover, Figure 7B can be moved to supplementary data eliminating the Ponceau panel and reducing the green background of pIKKA/B. I suggest adding a positive control if possible, for the Western Blots.

12) In the sentence from line 318 to 319, the authors described that the dCas9 can bind the DNA “promiscuously”, regarding this I didn’t see in all the experiments the usage of the dCas9 fused to activators alone, to evaluate this promiscuous behavior. Indeed, they showed only the association with trcrRNA (please clarify the identity of this construct).

13) It was previously described that lncRNA RP11 is associated with tumor differentiation [Ke Su et al. 2018, ISSN:2156-6976/ajcr0081106] and interacts in someway with IL-6 transcription. The authors should check the eventual interaction between the lncRNA with the IL-6 transcription and verify if there are more articles describing this phenomenon.

14) As a suggestion, after all the expression data using qPCR, I think it is more consistent to show an increment of the IL-6 protein. As matter of fact, it is clearly demonstrated that RNA increase is not directly associated with protein increment.

---

## [Author Response · Author response to Decision Letter 0]

10 Sep 2019

1. We would recommend that you include in the methods section the source of all the cell lines used in your study. Done as suggested.

We have revised accordingly. For 2 of 3 instances references to “Data not shown” have been replaced with in-text values, while for the third, the data are now included as a supplemental figure (S4 Fig).

Reviewers' comments:

Reviewer's Responses to Questions

5. Review Comments to the Author

Reviewer #1: Soubeyrand et al describe the finding of a CRISPRa off-target effect leading to increased IL6 expression while targeting a long non-coding RNA.

Major comments/suggestions

Line 99: There should be some explanation in the results section of the work leading to Figure 1. The relevant section has been expanded and the actual published work has now been referenced in order to facilitate understanding. 

Line 107: The authors should annotate the primers in the schematic representation of the locus. Furthermore, the authors need to explain how the results support the presence of an alternatively spliced exon.

The presence of an alternatively spliced exon is inferred from the systematic presence of a doublet using distinct non-overlapping primers spanning the same region spanning exon 1 and exon 2. The upper band is predicted to result from the presence of an additional sequence of about 100 bp, possibly of exonic origin. As the product was not sequenced, this is only inferred (hence the term “consistent”. Note that alternative splicing is quite prevalent in lncRNAs and retention of longer exons or inclusion of additional exons is common. Wording in the Fig 1 legend was modified to better convey this. In addition, to further facilitate understanding, the oligonucleotides have been annotated (numbered) as suggested. The last lane which was a technical repeat of an earlier lane, was removed.

Line 118: The authors should explain how the reporter construct is employed, and what it is.

This has now been clarified in the Results section.

Figure 3 Legend: The authors should explain the normalization strategy more clearly. I cannot understand what they mean by “Results are displayed as a function of the predicted binding region of the cgRNA, relative to exon 1 of RP11, set to 1”.

We agree. The sentence was confusing and the normalization strategy clarified. 

Line 127: If the region of maximal activation overlaps with the DNAseI hypersensitivity and ChIP Seq region, why is guide -267 and -102 not more active?

There could be numerous reasons for this, including occupancy by existing factors competing with CRISPR/Cas9 or positional/orientation constraints relative to the transcriptionstart site. We have now included a short sentence to the effect that accessibility is insufficient for activation.

Line 149: Describe the experiment “Array analysis”

This section has been reworded to clarify the experimental context.

Line 212: The authors state that only heterozygotes were found. How many clones were tested? Is there an explanation why there were only heterozygotes? Where the clones heterozygote for the ~400pb deletion? Also, show the editing profiles.

48 clones were tested and only three heterozygotes were obtained. This information has now been added to the methods section and clarified in the legend. This outcome is not uncommon as dual deletion are rare in CRISPR edition, probably as a result of the dominance of the non-homologous end joining which favors rapid editing and rejoining of ends at the expense of the desired longer distance recombination/rejoining events. Based on our findings we estimate that the likelihood of obtaining a knock-out (biallelic deletion of the promoter region) to approach 0.3% (3/48 X 3/48). Sanger sequencing results of the heterozygotes are included as Fig 6 (promoter/exon 1 deletion) and within S4 Fig (SG-505 editing).

Line 246: The authors should perform an unbiased off-target detection experiment, such as GUIDE-seq. This is not, as the authors state in line 323, a colossal undertaking. Furthermore, this can shed more light on the mechanism underlying the activation of IL6. I understand that multiple guides gave a similar phenotypic result (IL6 upregulation), but this could be explained by similarity in the promotor or enhancer regions.

We agree that colossal is a superlative here. The end of the sentence has been removed and changed to reflect our ongoing investigation into this locus. Experimental off-target detection methods such as GUIDE-seq offer a very useful way to interrogate active CRISPR/Cas9 specificity. Unfortunately, this approach does not address binding per se, but cleavage, which requires more stringent mechanistic requirements than mere binding. As such it is not suited for dCAS9 interrogation where cleavage is not involved. As discussed in the manuscript, by removing the DNA cleavage requirement, productive dCAS9 “events” are predicted to be much more promiscuous. Rather one would have to resort to genome-wide dCAS9 binding assays (ChiP style) in the presence and absence of cgRNA. As mentioned previously in the manuscript, this analysis has been performed, albeit in a different context (Wu et al, 2014). 

As for the IL6 link, an approach could be to use 2-3 cgRNA that activate IL6 and control ones. While we feel that such an experiment has indeed considerable appeal in that all/most binding events would be identified, the findings would require a series of follow-up experimental validation to disentangle IL6 relevant from IL6 irrelevant interactions. This question is most appropriate for a future publication. 

Figure 7A: How was the normalization performed?

Modifications have been made to the legend to clarify this point.

Minor comments/suggestions

Line 50: S.p.Cas9 does not require a single guideRNA (sgRNA) as it can also employ a two-part guide RNA consisting of a crRNA and a tracrRNA.

Agreed. This has been modified to reflect the native CRISPR-Cas9 situation.

Line 54: What do the authors mean by suboptimal cleavage? Off-target sites can lead to high editing levels even in the presence of mismatches.

Agreed. Suboptimal was removed.

Line 57: Maybe the authors can refer to the dCas9 that is being used in this study.

This was mentioned in the Methods (Addgene number). Mention to the differently humanized (i.e. synonymous but distinct codons) is now made. 

Line 60: What do the authors mean by “...whose efficiency reflects its nuclease ability…”

This sentence was largely redundant and was removed.

Figure S1: Explain CARMAT.

CARMAT is our in-house name for the transcript; this has been replaced with RP11, which was used throughout the manuscript. This is now corrected.

Figure 2 Legend: The DNAse rich region is defined as a black bar. However, there are 3 different black bars in the figure.

The black reference has been removed as it was confusing. Rather the reader is pointed to the bars legends on the left

Figure 3 Legend: Explain the following terms: PPIA & cgRNA.

Full gene name is now provided and cgRNA was changed to sgRNA which is used throughout the manuscript.

Line 177: Which list are you referring to as “nominally affected”?

This sentence has been clarified. 

Line 189: The section head suggests that the RP11 promotor drives IL6 activation. This seems a bit strong.

Clearly not our intention, as our data point to off-target effects. The title was changed to more appropriately reflect this. 

Line 209: Should “CRISPR/Cas9” be replaced by “CRISPRa”?

Thank you for bringing this to our attention. The sentence was modified to include both CRISPR systems and an incorrect mention of CRISPR/Cas9 was rectified.

Line 276: Can the authors describe the rationale of the experiments described in this paragraph?

The paragraph was reorganized with the rationale (lines 285-287) earlier in the paragraph. An additional sentence has been added to improve comprehension.

Reviewer #2: This is a manuscript about the possible off-targeting problems relative to the usage of the emerging CRISPR activation system. The authors described the possible interference of the CRISPRa with the IL-6 cytokine, that can generate some concerns for the usage of this tool in future therapies. Although the idea is interesting, in my opinion the study needs some adjustments and clarifications before publication:

1) Why did the authors choose RP11 and not TCONS that are in the same Risky area?

Because of their lack of obvious coding potential and lower conservation, determining the exonic structure of lncRNAs is somewhat challenging. RP11 and TCONS correspond to the same gene locus. TCONS_00023493 and RP11-326A19.4 are derived from semi-automated annotations of putative transcripts using different underlying data sets and integrations: RP11-326A19.4 is predicted by the GENCODE genes project and TCONS transcripts are predicted by the Human Body Map consortium. This is explained in the legend. 

2) Please, redesign Figure 1B because the scheme representing the primers binding is not clear and the results are not easy to read, make it clear. The PCR bands are too smeared.

As mentioned in the response to Reviewer 1, the figure has been simplified. As for the PCR bands being too smeared, this is in part due to the PCR buffer conditions (PCR Terra direct) and the small size of the fragments. Attempts were made to improve visualization by adjusting the exposure and contrast of the image. Two bands were reproducibly observed with exons spanning intron 1 (linking exon 1 and 2) as addressed in the figure legend.

3) Figure 2B: statistics analysis is missing. Please, explain why the MFGE8 promoter was chosen as control in this experiment.

All changes were significant relative to the control plasmid (promotorless). All promoters conferred similar inductions (not statistically different from each other, ANOVA). This is now clarified in the figure legend. MFGE8 was chosen as a logical progression of our previous works where we characterized the impact of the CARMA region (RP11 is located between MFGE8 and ABHD2). More relevant to this manuscript is the fact that MFGE8 is widely expressed at relatively high level, indicating that the promoter is quite potent; this information has been incorporated in the Results section.

4) Figure 3: y-axis the relative expression level is normalized to cells transfected with trcrRNA? This is not clear indeed in Figure 4A the y-axis name changes in “normalized to trcrRNA plasmid” that is not correct, because is it not possible to compare cDNA quantification with a plasmid. Moreover, it should be normalized using a housekeeping gene for the specific cell type.

Normalization was not performed vs plasmid DNA, but to values (X/PPIA) derived from cells transfected to dCAS9 + trcrRNA plasmids. The ordinate description has been changed to facilitate understanding (throughout the manuscript).

5) The paragraph from line 163 to 174 is not clear please try to explain in a different way.

Table 1 and its attached text have been expanded to address this. Over representation analysis is briefly explained.

6) In the text, lines 184-185 the authors stated that after CRISPRi (using dCas9 fused to KRAB inhibitor) the RP11 expression was 30% lower with no effects on IL-6. It seems strange because it is known (Gao, X., Tsang, J. C., Gaba, F., Wu, D., Lu, L., & Liu, P. (2014) Nucleic acids research, 42(20), e155-e155.) that CRISPRi is even more effective for endogenous genes, and the off-target effect is mainly due to sgRNA used. This result is not demonstrating an off-target of the system because the RP11 can be an enhancer of the IL-6 gene (already demonstrated for lncRNA) and repressing it doesn’t mean that the endogen IL-6 is not transcribed.

Rules for optimal CRISPRa and CRISPRi differ somewhat, with optimal CRISPRi typically needing sgRNA internal to the transcribed region (essentially steric hindrance of the pol II complex progression), whereas CRISPRa is more effective towards the upstream region to help engage the complex. In that sense, SG-286 is not optimal for CRISPRi, which likely accounts for the small inhibition on RP11. SG-286 was used in order to better compare both systems (similar off-targets etc…) A sentence has been added to this effect.

7) Figure 4B: the authors showed that using different sgRNAs on RP11 promoter is leading to an increase of IL-6 due to CRISPRa. This seems real if consider only sgRNA -567, -505 and -286 (the guide with the strongest effect) but my concerned are about sgRNA -546, indeed using that guide the increase of RP11 seem to be not significant and the associated IL-6 is very high, can the authors speculate a little on this result? Moreover, if the off-target effect is not driven by sgRNA as explained with this figure, how can the deactivated Cas9 binds the DNA to drive the transcriptional machinery?

These inconsistencies were duly noted in our previous version but not explained. These findings were interpreted as possible off-target effects on IL6 and this is now explicitly mentioned. 

.

8) Figure 5: the authors tried to generate a partial promoter knock-out part of the 5’ UTR and exon 1, and they failed, “Unfortunately, the process yielded only compound heterozygotes”. My concerns regarding these experiments are due to the deletions: in three out of four deletions is resulted in heterozygotes (also if is not so clear, the smaller band seem two and not only one) and in the first one failed. Which cells did the authors use to perform the deletions? Maybe the problem is due to the presence of more than two copies of the genomic asset of the cell line? That should be clarified because the effect resulted can be also influenced by that. Moreover, I will suggest adding a chromatogram showing the junctions of the deletion, in fact using two sgRNAs is leading to a non-homologous recombination that can also interfere with the sgRNA -505 binding site. In Figure 5C the relative expression of RP11 and IL-6 in the wild type are opposed to what the authors showed in figure 4B, in Figure 5C the levels of RP11 in wild type are higher than IL-6 after transfection with both sgRNA-286 and -505. It diminishes the consistence of the data on heterozygous, thus the increment of IL-6 seem to be related to cells external or internal conditions.

Sequences over the edited regions are now included, for both SG-505 deletion and RP11 deletion, as mentioned in the response to reviewer 1’s comments. Cells used (as mentioned in the methods and now explicitly mentioned in the results section) were HEK293T. Copy number values over the RP11 region are normal (diploid) according to ENCODE/HAIB HEK293 data (track accessible through the UCSC browser hg19 build). Expression findings (loss of RP11 expression in heterozygotes) are also consistent with a de facto KO of the transcript. As for variability, indeed we did observe that the absolute fold value changes did vary somewhat across sets of experiments (typically performed in sets of 3), perhaps reflecting particular experimental noise (basal expression of IL6 and RP11, cell density variability etc..). Still, the overall trend is strongly consistent: the relative ability to induce IL6 and RP11 by the various sgRNA.

9) In the Figure 6 is not clear why the authors did choose the SRP14 as a control. Moreover, they changed the relative for the normalization from transfected with the trcrRNA to PPIA. It would be more linear If they normalize everything to a fixed housekeeping. In addition, the standard deviation of the RP11 values at 72h is very high, maybe the experiment should be confirmed.

Our laboratory has observed that SRP14 expression is relatively stable and is often used as a housekeeper gene by various groups. We feel its inclusion further emphasizes the (seemingly) unique link between IL6 and RP11 upregulations. The experiment has been repeated (for n=3) and is now normalized to control as suggested. This results in a smaller error bar largely due to within experiment corrections rather than the additional repeat (S.D. are not dependent on sample size, only S.E. are). .

10) Did the authors try the alignment of the IL-6 and the RP11 promoters? I will advise doing it to understand the dissimilarity between them. 

Yes, as mentioned in the Discussion, extensive similarity searches between IL-6 and RP11 genes were performed but revealed no significant identity (search was done by aligning sgRNA sequences with the IL6 gene). 

11) In Figure 7A, the authors want to demonstrate the inhibition effect of IL-6 transcription marking different kinases? These experiments are not clear and need to be represented in a better way. Moreover, Figure 7B can be moved to supplementary data eliminating the Ponceau panel and reducing the green background of pIKKA/B. I suggest adding a positive control if possible, for the Western Blots.

Changes have been made to facilitate comprehension. First, the figure was split in 2 distinct figures. Second, the SRP14 controls were moved to the Supplementary section. Third, additional in text clarifications were introduced in the results and figure legend.

In figure 7B (now 9), we have elected to leave the Ponceau stains as is as we feel that loading controls are important in Western blot experiments. As suggested we now include a positive control (S7 Fig) for pIkka/b performed with the same tube of Antibody and THP-1 cells, which are known to undergo increased Pikka/b during polarization into pro-inflammatory M1 stage (https://www.cellsignal.com/products/primary-antibodies/phospho-ikka-b-ser176-180-16a6-rabbit-mab/2697).

12) In the sentence from line 318 to 319, the authors described that the dCas9 can bind the DNA “promiscuously”, regarding this I didn’t see in all the experiments the usage of the dCas9 fused to activators alone, to evaluate this promiscuous behavior. Indeed, they showed only the association with trcrRNA (please clarify the identity of this construct)

The reviewer is correct, in that the promiscuous behaviour is suggested only for the trcrRNA. Binding in the absence of any RNA cofactor is possibly lower but would not as adequately control for the scaffold RNA.

13) It was previously described that lncRNA RP11 is associated with tumor differentiation [Ke Su et al. 2018, ISSN:2156-6976/ajcr0081106] and interacts in someway with IL-6 transcription. The authors should check the eventual interaction between the lncRNA with the IL-6 transcription and verify if there are more articles describing this phenomenon.

RP11-443P15.2 alluded to in the above publication is unrelated to the RP11-326A19.4 described in this manuscript. RP11 is only used for simplification. The RP11 naming scheme covers thousands of unrelated speculative transcripts from EncodeGencodeBasicV28. To dispel any possible confusion targeted references to RP11 in the Discussion part have been modified to RP11-326A19.4.

14) As a suggestion, after all the expression data using qPCR, I think it is more consistent to show an increment of the IL-6 protein. As matter of fact, it is clearly demonstrated that RNA increase is not directly associated with protein increment.

We agree and have now included ELISA data demonstrating increased IL6 in the supernatant of HEK293T in response to 48 h of CRISPRa (Fig 4B). 

---

## [Decision Letter · Decision Letter 1]

7 Oct 2019

Off-target effects of CRISPRa on interleukin-6 expression

PONE-D-19-18163R1

Dear Dr. Soubeyrand,

We are pleased to inform you that your manuscript has been judged scientifically suitable for publication and will be formally accepted for publication once it complies with all outstanding technical requirements.

With kind regards,

Serif Senturk, Phd

Academic Editor

PLOS ONE

Reviewers' comments:

Reviewer's Responses to Questions

**Comments to the Author**

1. If the authors have adequately addressed your comments raised in a previous round of review and you feel that this manuscript is now acceptable for publication, you may indicate that here to bypass the “Comments to the Author” section, enter your conflict of interest statement in the “Confidential to Editor” section, and submit your "Accept" recommendation.

Reviewer #1: All comments have been addressed

2. Is the manuscript technically sound, and do the data support the conclusions?

Reviewer #1: Yes

3. Has the statistical analysis been performed appropriately and rigorously? 

Reviewer #1: Yes

4. Have the authors made all data underlying the findings in their manuscript fully available?

Reviewer #1: Yes

5. Is the manuscript presented in an intelligible fashion and written in standard English?

Reviewer #1: Yes

6. Review Comments to the Author

Reviewer #1: The authors have addressed the comments/suggestions adequately. There are a few small corrections necessary, which are outlined below:

Line 63: CRISPRr should be CRISPRi

Line 244: the deletion described can only be done with CRISPR/Cas9, not with CRISPRa

Line 446: abscissa is more of a mathematical term, maybe refer to ‘x-axis’

Line 450: instead of ‘empty sgRNA’, the authors can describe the negative control as ‘tracrRNA only’

7. PLOS authors have the option to publish the peer review history of their article (what does this mean?). If published, this will include your full peer review and any attached files.

Reviewer #1: No

---

## [Editor Report · Acceptance letter]

15 Oct 2019

PONE-D-19-18163R1 

Off-target effects of CRISPRa on interleukin-6 expression 

Dear Dr. Soubeyrand:

I am pleased to inform you that your manuscript has been deemed suitable for publication in PLOS ONE. Congratulations! Your manuscript is now with our production department. 

With kind regards,

on behalf of

Dr. Serif Senturk 

Academic Editor

PLOS ONE